

# pyPI (v1.3): Tropical Cyclone Potential Intensity Calculations in Python

Daniel M. Gilford[1,2]

[1]Institute of Earth, Ocean, and Atmospheric Sciences, Rutgers University, 71 Dudley Road, Suite 205, New Brunswick, NJ 08901, USA.
[2]Department of Earth and Planetary Sciences, Rutgers University, Piscataway, NJ, USA.

**Correspondence:** Daniel M. Gilford (daniel.gilford@rutgers.edu)

**Abstract.** Potential intensity (PI) is the maximum speed limit of a tropical cyclone found by modeling the storm as a thermal heat engine. Because there are significant correlations between PI and actual storm wind speeds, PI is a useful diagnostic for evaluating or predicting tropical cyclone intensity climatology and variability. Previous studies have calculated PI given a set of atmospheric and oceanographic conditions, but although a PI algorithm—originally developed by Kerry Emanuel—is in

widespread use, it remains under-documented. The Tropical Cyclone Potential Intensity Calculations in Python (pyPI, v1.3) package develops the PI algorithm in Python, and for the first time details the full background and algorithm (line-by-line) used to compute tropical cyclone potential intensity constrained by thermodynamics. The pyPI package (1) provides a freely available, flexible, validated Python PI algorithm, (2) carefully documents the PI algorithm and its Python implementation, and (3) demonstrates and encourages the use of PI theory in tropical cyclone analyses. Validation shows pyPI output is nearly

identical to the previous potential intensity computation, but is an improvement on the algorithm's consistency and handling of missing data. Example calculations with reanalyses data demonstrate pyPI's usefulness in climatological and meteorological research. Planned future improvements will improve on pyPI's assumptions, flexibility, and range of applications and tropical cyclone thermodynamic calculations.

# 1 Introduction

Tropical cyclones pose significant risks to coastal societies, being among the costliest and deadliest of global natural hazards (e.g. Pielke et al., 2008; Rappaport, 2014; Hsiang and Jina, 2014). Damages increase exponentially with tropical cyclone intensity ($\sim$5% per ms$^{-1}$ Murnane and Elsner, 2012), so it is crucial to understand and accurately bound tropical cyclone maximum wind speeds. Theoretical and numerical models (e.g. Emanuel, 1987; Tsuboki et al., 2015; Sobel et al., 2016; Wehner

et al., 2018) along with recent observations (Kossin et al., 2020) indicate that climate change has already increased storm intensities—a trend expected to continue as the earth system warms. Emanuel (2005) showed the total destructive potential of tropical cyclones (derived from time-integrated maximum intensity) has increased since the 1970s. Elsner et al. (2008) showed





that the most intense observed tropical cyclones are getting stronger, and a more recent comprehensive study shows that the number of major (Category 3+) tropical cyclones has increased over the past 40 years (Kossin et al., 2020). Given the links
between intensity and tropical cyclone impacts, it is worthwhile to develop and improve modeling tools for diagnosing and predicting tropical cyclone intensities.

Potential intensity (PI) is a theoretical model for the upper bound (colloquially known as the "speed limit") on tropical cyclone intensity, given environmental conditions and energetic constraints (e.g. Emanuel, 1986; Holland, 1997). PI has several properties which make it a particularly useful model for studying tropical cyclones. First, PI is statistically linked to the lifetime
maximum intensities of observed storms (Emanuel, 2000), so it can be used to assess and interpret real-world intensity trends and variability (e.g. Wing et al., 2007; Gilford et al., 2019; Shields et al., 2019). Second, PI can be readily calculated from standard atmospheric profiles (either modeled or observed), making it flexible across many applications and spatio-temporal scales. Third, PI may be decomposed into thermodynamic and parametric contributions that enable budget and sensitivity analyses—with direct implications for real-world storms. Finally, as a theoretical model grounded in meteorological data, it
is well-suited for incorporation into prognostic and diagnostic indices of intensification (e.g. the ventilation index, VI, Tang and Emanuel, 2012), tropical cyclogenesis (e.g. VI; the genesis potential index, GPI, Camargo et al. 2007; the tropical cyclone genesis index, TCGI, Tippett et al. 2011), and destructive potential (e.g. the power dissipation index, PDI, Emanuel, 2005).

The algorithm to compute PI was original developed by Bister and Emanuel (2002) (hereafter BE02), coded as a FORTRAN subroutine. It was later converted for use as a MATLAB function by Kerry Emanuel, and has been irregularly revised by Kerry
and other collaborators/colleagues; the most up-to-date version of the MATLAB function is available online at ftp://texmex. mit.edu/pub/emanuel/TCMAX. The BE02 PI function has been extensively (and nearly-universally) used and/or adapted by the tropical meteorology community to calculate PI for modeling, observational, and theoretical research applications (e.g. McTaggart-Cowan et al., 2008; Gualdi et al., 2008; Camargo et al., 2009; Sobel and Camargo, 2011; Tippett et al., 2011; Bryan, 2012; Vecchi et al., 2013; Ramsay, 2013; Camargo, 2013; Chavas and Emanuel, 2014; Strazzo et al., 2014a, b; Wing
et al., 2015; Sobel et al., 2016; Polvani et al., 2016; Lin and Emanuel, 2016; Gilford et al., 2017; Xu et al., 2019; Gilford et al., 2019; Emanuel, 2018; Shields et al., 2019; Camargo and Polvani, 2019, and many others). The BE02 function is also used to compute daily maps of North Atlantic PI for meteorological assessment in real-time (produced by the Center for Land–Atmosphere Prediction and available online at http://wxmaps.org/pix/hurpot, Emanuel et al. 2004).

Despite widespread use, the BE02 algorithm itself has never (to my knowledge) been fully documented. Because it is an
important modeling tool for tropical cyclone intensity, there is a need for transparent and documented PI algorithm. It is also advantageous to implement a PI algorithm in Python (which is freely available and has many advantages in scientific research, Millman and Aivazis 2011), to complement the existing counterparts in MATLAB (which is proprietary and therefore less accessible) and FORTRAN (which is not easily extensible for a broad range of applications, Rashed and Ahsan 2018).

I developed Tropical Cyclone Potential Intensity Calculations in Python (i.e. "pyPI") to meet these needs. In addition to
adapting the BE02 algorithm in Python and thoroughly documenting the model, pyPI provides a maintained and regularly archived repository to support open science in the tropical meteorological community. pyPI is also ideally suited for ongoing community development and improvement, and for research applications which require flexibility in particular PI input





parameters or components (for example, the computation of the lifting condensation level) or integration with other Python packages. This manuscript provides context for the initial package release of pyPI (v1.3) and details its development, algorithm, validation, and sample applications.

The proceeding Sect. 2 provides a brief overview of potential intensity theory and introduces its key components including thermodynamic efficiency and disequilibrium. Section 3 presents the mathematical basis of pyPI's potential intensity computations. I describe the Python implementation of the pyPI algorithm in Sect. 4, including its adjustable input parameters and handling of missing data. Model validation in Sect. 5 demonstrates that pyPI output is nearly identical to the previously published MATLAB algorithm, with minor improvements for consistency. Section 6 illustrates several climatological applications of pyPI. I conclude with a discussion of planned pyPI advancements in Sect. 7.

## 2 Potential Intensity Theory

Tropical cyclones arise in response to a thermodynamic gap in the tropical atmosphere's energy budget (e.g. Emanuel, 2006). The tropical surface's output longwave radiative cooling is outpaced by combined solar and longwave radiative heating (terrestrially sourced by greenhouse gases and clouds) received at the surface. In the absence of any balancing outgoing process, the resulting thermodynamic disequilibrium would lead to a build-up of heat driving substantially higher surface temperatures (e.g. Manabe and Strickler, 1964). Instead, atmospheric convection plays the leading role in removing this excess heat; tropical cyclones are a well-known expression of this convection.

Driven by thermodynamic disequilibrium—which is largest in the summer and autumn seasons—an existing mature tropical cycle will transfer heat from the surface to the atmospheric boundary layer, largely through latent heat release of evaporation and from the sea surface and dissipative heating (Bister and Emanuel, 1998). Viewed from this perspective, it is useful and convenient to model tropical cyclones as Carnot heat engines (Emanuel, 1987) which convert this fuel (i.e. thermodynamic disequilibrium) to kinetic energy in the form of azimuthal winds. Figure 1 shows a diagram of a Carnot cycle overlaid on a cross-section of a mature tropical cyclone, along with the pyPI algorithm inputs and outputs.

Following the entropy gradient, air at the outer reaches of the storm spirals inward (branch A) toward the minimum central pressure in the eye ($p_{min}$) and the entropy maximum near the radius of maximum winds (RMW). Along its motion this air gathers entropy through isothermal heat absorption (through the two processes noted above) with the temperature of the sea surface, $T_s$. When the air reaches an entropy maximum at the RMW it bends upward through adiabatic expansion (branch B), conserving its entropy as it rises through the eyewall and then along the outflow at the storm top. This outflow layer is called the "outflow temperature level" (OTL), and here the air undergoes isothermal radiative heat loss (branch C) with temperature $T_0$, transferring the entropy generated by the storm to its surroundings. Finally, the Carnot cycle closes as the air undergoes adiabatic compression with lower entropy back towards the sea surface (branch D) while its temperature rises once again.

An advantage of this theoretical model of a tropical cyclone is that it permits a formulation of the storm's theoretical maximum intensity—i.e. its PI—in terms of the heat engine efficiency, defined by the temperatures at each extent (reservoir)





## The Carnot Cycle and Potential Intensity

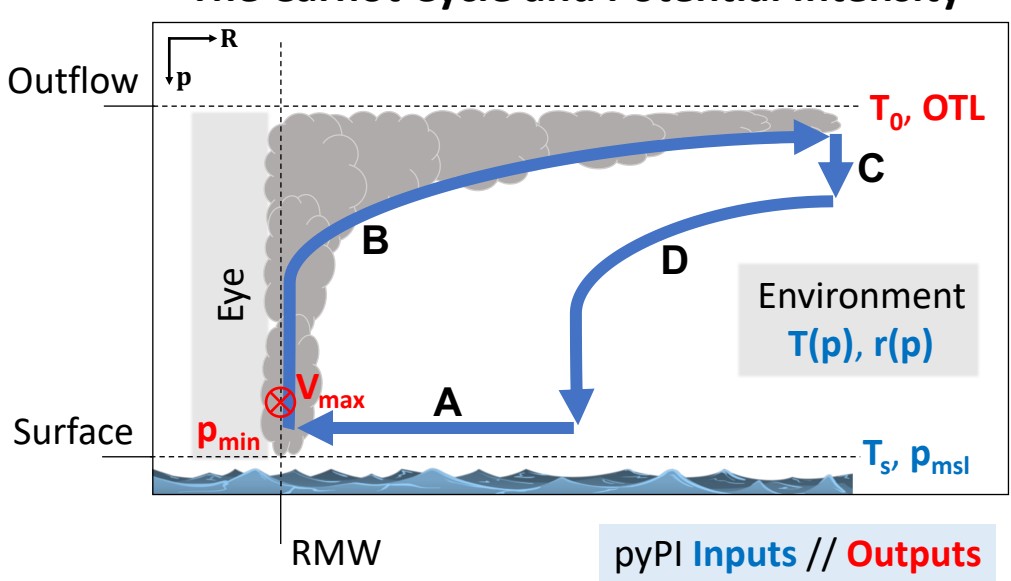

**Figure 1.** The cross-section (along radius, $R$, and pressure, $p$) of an idealized and mature tropical cyclone and its thermodynamic cycle. pyPI inputs and outputs are in blue and red text, respectively, and are defined in Table 1. Bold blue lines and black letters indicate the four branches of the Carnot cycle, A through D (see text). The tropical cyclone maximum potential intensity ($V_{max}$) is found at the radius of maximum winds (RMW) and is directed into-the-page in the northern hemisphere. The minimum central pressure ($p_{min}$) is found in the storm's eye. Based on Carnot cycle illustrations in Emanuel (1987, 2006).

of the engine ($\frac{(T_s - T_0)}{T_0}$), and in terms of the heat source itself (i.e. thermodynamic disequilibrium). These quantities may be estimated with atmospheric and oceanic observations (Sect. 5.1).

As derived in Bister and Emanuel (1998) and Emanuel (2003), the maximum (near-surface) potential intensity of a tropical cyclone, $V_{max}$, may be approximated by:

$$(V_{max})^2 = \frac{C_k}{C_D} \frac{(T_s - T_0)}{T_0} (h_o^* - h^*) \tag{1}$$

where $C_k$ and $C_D$ are the enthalpy and momentum surface exchange coefficients, respectively, $h_o^*$ is the saturation moist static energy at the sea surface, and $h^*$ is the saturation moist static energy of the air above the boundary layer (often evaluated at $\sim$500–600 hPa, cf. Wing et al. 2015). Tropical cyclone thermodynamic disequilibrium and efficiency are represented by the terms ($h_o^* - h^*$) and $\frac{(T_s - T_0)}{T_0}$, respectively; the ratio $\frac{C_k}{C_D}$ is a defined constant that may be estimated from theory or observations (Sect. 4.1).

In physically-based axisymmetric models, mature tropical cyclone wind speeds tend to reach their PI (e.g. Rotunno and Emanuel, 1987). In contrast, observed storms rarely attain their thermodynamically-constrained potential intensities (see lim-



itations discussed in Sect. 7). However, Emanuel (2000) combined climatologically-derived PI with observed tracks and intensities of real-world storms to show that any observed storm statistically has an equal likelihood of attaining any lifetime maximum speed between some lower bound (symbolized as "$\alpha$" in Gilford et al., 2019) and the PI along its track. This is a

powerful statistical property, because it implies that any shift in the PI distribution—either on short timescales such as during an anomalously cold summer or on long timescales such as a response to a warming climate—will be accompanied by similar shifts in the observed intensity distribution (cf. Wing et al., 2007). Gilford et al. (2019) showed that randomly sampled observed intensity distributions which have at least 25 tropical cyclones (of hurricane strength or greater) will robustly follow their associated along-track potential intensity distributions.

Links between observed and potential intensities have been shown on seasonal (Tonkin et al., 2000; Gilford et al., 2019), interannual (Wing et al., 2007; Shields et al., 2019), and climatological (e.g. Emanuel, 2000) timescales. The relationship is more robust when PI is evaluated along the track of a storm rather than as a basin-wide average (Wing et al., 2007; Gilford et al., 2019; Shields et al., 2019). Other studies have examined the roles of volcanic eruptions/lower stratospheric variability (e.g. Emanuel et al., 2013), the Montreal protocol (Polvani et al., 2016), or climate change on potential intensity (Emanuel,

2005; Vecchi et al., 2014; Sobel et al., 2016). Any oceanic or atmospheric variability or trend which alters the thermodynamic environments of tropical cyclones could have some effect on PI (though the relative importance and/or statistical significance of these effects will vary). The connection between tropical cyclone PI and climate change will likely remain a critical topic to understand: the troposphere continues to warm and moisten in response to anthropogenic emissions of greenhouse gases, while simultaneously the lower stratosphere is cooling (e.g. IPCC, 2013; Santer et al., 2013).

## 3 The pyPI Algorithm

### 3.1 BE02 PI Formulation

This section provides a detailed description and record (including relevant citations) of the PI algorithm with its thermodynamic/meteorological origins, assumptions, and computations. In addition to outlining the numerical computation here, pyPI's algorithm module has been commented for reference (Gilford, 2020).

Potential intensity may be derived following Bister and Emanuel (2002) (which is largely based on the formulation of Emanuel 1995) idealizing a tropical cyclone as a Carnot heat engine (e.g. Emanuel, 1987), and assuming (1) the work done against friction by the outflow is ignored, (2) when the storm intensity reaches its maximum the anticyclone at the top of the storm is fully developed, and (3) the gradient wind may be approximated by cyclostrophic wind at the RMW. Under these conditions, the Carnot cycle formulation yields an expression for the maximum potential intensity (which roughly scales with

the approximated PI expression, Eq. 1; Gilford et al. 2017; Wing et al. 2015):

$$(V_{max})^2 = \frac{T_s}{T_0} \frac{C_k}{C_D} (CAPE^* - CAPE_{env})|_{RMW} \qquad (2)$$





---

**Algorithm 1** pyPI's iterative procedure to calculate tropical cyclone potential intensity.

---

**input:** $T_s, p_{msl}, p, T(p), r(p)$

1: Check that inputs are appropriate

2: Calculate $CAPE_{env}$

**repeat**

    3: Calculate $CAPE|_{RMW}$

    4: Calculate $CAPE^*$, OTL, $T_0$

    5: Estimate $p_m$ at this $i^{th}$ iteration

**until** convergence. Objective: $\Delta_{i,i-1} p_m < 0.5$ hPa

6: Calculate final $p_{min}$

7: Calculate $V_{max}$

**return** $V_{max}, p_{min}$, OTL, $T_0$, algorithm flag

---

where $CAPE^*$ is the convective available potential energy of saturated air lifted from sea level to the outflow level referencing the environmental profile, and $CAPE_{env}$ is the convective available potential energy of the environment. Because the final term is evaluated at the RMW and $CAPE^*$ is pressure dependant, an expression for the surface pressure at the RMW is needed. Following Bister and Emanuel (2002) (cf. also Garner 2015, their Eq. 6), the minimum pressure of the tropical cyclone at the RMW, $p_m$ is found with[1]:

$$R_d T_v log(\frac{p_{msl}}{p_m}) = \frac{1}{2}(V_{max})^2 + CAPE|_{RMW}, \tag{3}$$

where $T_v$ is the surface environmental virtual temperature, and $CAPE|_{RMW}$ is the environmental convective available potential energy evaluated at the RMW. Because the boundary layer water vapor mixing ratio is higher in the tropical cyclone eyewall than the storm's outer region (assuming a constant relative humidity in the boundary layer across the storm's radius), $CAPE|_{RMW}$ is slightly larger than $CAPE_{env}$ (discussed more below).

The pressure dependence of $CAPE$ requires solving Eq. 2–3 with numerical iteration, which pyPI performs with individual PI and $CAPE$ modules. Algorithm 1 summarizes how pyPI computes maximum potential intensity by modeling a tropical cyclone as a Carnot heat engine. Algorithm inputs/outputs are provided in Table 1 and described in Sect. 4; meteorological constants are provided in the Appendix (Table A1). I begin by describing the $CAPE$ calculation, which is used throughout the pyPI algorithm.

## 3.2 $CAPE$ Module

$CAPE$ is defined as the sum of positive and negative areas of buoyancy energy of a lifted parcel on a sounding (e.g. Emanuel 1994, their Eq. 6.3.6, discussed more below) and is calculated by pyPI with the procedure in Algorithm 2.

---

[1]Eq. 4 in Bister and Emanuel (2002) mistakenly replaces $R_d$ with $c_p$. pyPI includes the correct factor of $R_d$.



---

**Algorithm 2** pyPI's procedure to calculate convective available potential energy.

---

**input:** $T_{j=0}, r_{j=0}, p_{j=0}, T(p), r(p), p$

1: Calculate parcel $s$

2: Find $p_{LCL}$

**for** each $j^{th}$ pressure level **do**

    **if** $p_j \geq p_{LCL}$ **then**

        3: Calculate $(T_\rho - T_{\rho,env})$ lifting the parcel along a dry adiabat

    **else if** $p_j < p_{LCL}$ **then**

        **repeat**

            4: Solve for $T_j$ at this $k^{th}$ iteration, conserving $s$

        **until** convergence. Objective: $\Delta_{k,k-1}T < 0.001$ K

        5: Calculate final $r_j$ dependent on the `ascent_flag`

        6: Calculate $(T_\rho - T_{\rho,env})$ lifting the parcel along a moist adiabat

    **end if**

**end for**

7: Find LNB, $T_{LNB}$

8: Calculate $CAPE$

**return** $CAPE, T_{LNB}$, LNB, algorithm flag

---

Given an initial surface parcel temperature, pressure, and mixing ratio, the procedure begins by finding the parcel's reversible entropy, which is conserved as it is lifted on the sounding. The parcel's water vapor pressure is found via the Ideal Gas Law (e.g. Bolton 1980, their Eq. 16):

$$e = \frac{r * p}{\epsilon + r}. \tag{4}$$

Saturation vapor pressure (in hPa) is given empirically as a function of $T$ in °C by Bolton (1980), their Eq. 10, following the
Clausius-Clapeyron relation:

$$e_s(T) = 6.112 * exp(\frac{17.67 * T}{T + 243.5}). \tag{5}$$

Then fractional relative humidity is defined as $RH \equiv \frac{e}{e_s} \leq 1.0$. Assuming the temperature dependence of specific heats is negligible over the range of temperatures in the tropical atmosphere, and integrating Kirchoff's equation (e.g. Emanuel 1994, Eq. 4.4.3–4.4.4) then the temperature dependence of the latent heat of vaporization is:

$L_v = L_{v0} + (c_{pv} - c_\ell) * T,$            (6)





with $T$ in °C. Finally, we are equipped to calculate the parcel's reversible total specific entropy (per unit mass of dry air), $s$, which is conserved as the parcel is lifted along the sounding (Emanuel 1994, their Eq. 4.5.9):

$$s = (c_{pd} + r_T c_\ell)log(T) - R_d log(p) + \frac{L_v r}{T} - r R_v log(RH),\qquad(7)$$

where $r_T$ is the total water content mixing ratio, which is identical to the parcel mixing ratio at the surface.

Having determined the parcel's initial moisture properties, we next find the lifting condensation level (LCL) of the parcel, in order to partition the upcoming buoyancy calculation between saturated and unsaturated regions of the profile. The pressure of the LCL, $p_{LCL}$, is found empirically with[2]:

$$p_{LCL} = p * RH^{(\frac{T}{A - B*RH - T})},\qquad(8)$$

where $A = 1669$ and $B = 122$. Note that the LCL of a lifted parcel that is already saturated is identical to its original pressure
level, i.e. $p_{LCL} \equiv p$. Likewise, parcels at levels below the LCL are (by definition) not saturated. After finding $p_{LCL}$ the CAPE algorithm begins an "updraft loop", where the positive and negative buoyancy of the parcel is calculated at every $j^{th}$ pressure level below the upper boundary on pressure (`ptop`, Sect. 4.1).

Starting with calculations at levels *below* the LCL ($p_j > p_{LCL}$), at each $j^{th}$ level the algorithm calculates the unsaturated parcel temperature by following a dry adiabat with the same temperature as the surface parcel. Applying Poisson's equation:

$$T_j = T * (\frac{p_j}{p})^{\frac{R_d}{c_{pd}}}.\qquad(9)$$

Because $CAPE$ is proportional to the positive and negative areas enclosed by the environmental and lifted parcel density temperatures ($T_{\rho,e}$ and $T_\rho$, respectively), we calculate the density temperature as (Emanuel 1994, their Eq. 4.3.6):

$$T_\rho = T * (\frac{1 + r/\epsilon}{1 + r_T}),\qquad(10)$$

where the net water mixing ratio ($r_T$) is the same as the parcel water mixing ratio at the surface, and in the environment below
the LCL before condensation has occurred (i.e. $r_T = r$ and $r_{T,j} = r_{j<LCL}$).

Next the algorithm finds the density temperature differences for all levels *above* the LCL ($p_j < p_{LCL}$). Because the parcel is saturated above the LCL, its moisture characteristics and temperature must be found iteratively at each $j^{th}$ level. First the algorithm solves for $r$ by rearranging Eq. 4, then until the numerical iteration converges (with objective for the parcel temperature: $\Delta T < 0.001$ K) it solves for parcel moisture characteristics which conserve the parcel's specific entropy $s$ (Eq.

---

[2]This is likely derived empirically from Bolton (1980), and was developed for Emanuel (1994) (Kerry Emanuel, *personal communication*). Modern calculations of $p_{LCL}$ are made following exact expressions from Romps (2017). A goal for future pyPI development is to replace the current empirical estimate with the modern formulation.




7) following a moist adiabat; finding these permits an estimation of the density temperature differences at each level. At the beginning of the loop, $T$ and $r$ are set equal to the previous iteration's findings, then the loop steps forward updating the parcel's temperature (and the dependant $L_v$ and water vapor mixing ratios) assuming $s$ is conserved following saturated reversible adiabatic displacement. Following Newton's method ($T_{n+1} = T_n + \frac{s(T_n)}{ds(T_n)/dT}$; Wallis 1685), when the difference between $s$ and this iteration's entropy, $s_k$, scaled by the rate of change of entropy with temperature, $s_\ell$, is small (i.e. $\frac{s-s_k}{s_\ell} < \texttt{AP} * 0.001$),

then the algorithm will converge to estimate the parcel temperature at this level, $T_j$. Here $\texttt{AP}$ is a numerical step size employed to speed convergence, which changes dynamically depending upon the number of iterations that have taken place. If at any given level the total number of iterations exceeds 500 (an excessive number of iterations), or if the water vapor pressure becomes unrealistically close to the level pressure, then the algorithm fails to converge and returns zero $CAPE$.

When the algorithm converges for a level, the final parcel mixing ratio is set depending on the ascent type chosen by the

user (Sect. 4.1). For pseudoadiabatic ascent (`ascent_flag= 1`), liquid water condensed in the parcel during its ascent is assumed to drop out of the parcel, such that the heat capacity of liquid water is neglected and the mixing ratio is a function of the final level temperature (i.e. $r_j = r(T_j)$). For reversible ascent (`ascent_flag= 0`) the total water (and its heat capacity) is retained following the parcel (i.e. $r_j = r_{j=0} = r_T$). For intermediate fractions of `ascent_flag`, the mixing ratio scales over $r(T_j) \rightarrow r_T$.

Note that the density temperature difference (and hence a parcel's buoyancy) with height is not strictly higher under either ascent assumption. Parcels lifted reversibly are always warmer than those lifted psuedoadiabatically, but the weight of the carried condensate also means these parcels are more dense until they reach the upper troposphere (Emanuel 1994, their Table 4.2). Accordingly, Gilford et al. (2017) found that psuedoadiabatic (typically more buoyant) PI calculations generally have higher altitude OTLs than reversible (typically less buoyant) PI calculations on monthly timescales.

Having determined $r_j$, the algorithm computes the density temperature for the parcel and the environment (Eq. 10), and calculates each level's density temperature differences, $T_\rho - T_{\rho,env}$. We are now equipped to calculate the lifted parcel's convective available potential energy. $CAPE$ is given by the vertically integrated buoyant energy between the level from which the parcel is initially lifted ($j = 0$) and the level of neutral buoyancy (LNB; $j = LNB$). Following Emanuel (1994), their Eq. 6.3.6:

$$CAPE = PA - NA, \tag{11}$$

where

$$NA \equiv - \int_{p_{j=LFC}}^{p_{j=0}} R_d(T_\rho - T_{\rho,env})dlog(p) \tag{12}$$

$$PA \equiv + \int_{p_{j=LNB}}^{p_{j=LFC}} R_d(T_\rho - T_{\rho,env})dlog(p). \tag{13}$$





Negative areas ($NA$) are vertical regions of negative buoyancy which inhibit spontaneous convection in the profile; positive areas ($PA$) are vertical regions of positive buoyancy which cause the parcel to rise assuming an initial upward displacement. Note that $CAPE$ is not defined for parcels without positive areas. By definition, the level of free convection (LFC) separates regions that are negatively buoyant (below) from regions that are positively buoyant (above). When $(j = LFC) > (j = LCL)$, then regions of the profile above the LCL and below the LFC may still be negatively buoyant.

The $CAPE$ algorithm numerically solves Eqs. 11–13 in five steps:

First, we find the maximum level of positive buoyancy (INB), i.e. the highest altitude $j^{th}$ level where $T_\rho - T_{\rho,e} > 0$. If this highest level remains at $j = 0$, then there are no positively buoyant levels and the function returns zero $CAPE$.

Second, noting that $dp/(p_{mean,j:j+1}) \approx dlog(p)_{j:j+1}$ at each layer over the levels $j$ and $j + 1$—where the average pressure of each layer is $p_{mean,j:j+1} = \frac{1}{2}(p_j + p_{j+1})$—we find the positive and negative areas between the second-highest altitude level 225 (i.e. $j = 1$) and the the maximum level of positive buoyancy.

Third, we find the residual negative area (if $j = LFC > 0$) or positive area (if $j = LFC = 0$) of the mean layer composed of the surface and the lowest level.

Fourth, we find the LNB and the temperature at the LNB, $T_{LNB}$, along with the residual positive area of the mean layer between the the maximum level of positive buoyancy and the LNB. If the INB is found at the highest valid level (constrained 230 by ptop, Sect. 4.1), then the LNB and $T_{LNB}$ are set at that level.

Finally, the negative and positive areas are added together with the residuals following Eq. 11. After this last step, the algorithm flag is set to indicate the algorithm has successfully computed $CAPE$. Then the values of $CAPE$, $T_{LNB}$, the LNB, and the flag are returned to the PI module.

### 3.3 PI Module

The PI module begins by checking to ensure that the input atmospheric profile is appropriate for the PI calculation. If not, missing values are returned by the algorithm (see Sect. 4.2). Water vapor mixing ratios above the boundary layer do not influence the PI calculation (they are redundant in $CAPE^* - CAPE_{env}$), so any missing $r$ values above the surface are replaced with 0 g/g.

Following Algorithm 2 described above, pyPI computes $CAPE_{env}$, assuming the environmental air parcel is lifted from the 240 lowermost input level in $p$.

Next, pyPI iteratively solves Eqs. 2 and 3 (with objective for the minimum pressure at the RMW: $\Delta_{i,i-1}p_m < 0.5$ hPa). The algorithm begins by calculating the convective available potential energy (computing Algorithm 2 at each $i^{th}$ iteration) iterating from the initial lowest-level environment inward toward the radius of maximum winds, $CAPE|_{RMW}$. At each iteration the mixing ratio is updated to account for pressure dependence ($r$ increases slightly as $p \to p_m$ approaching the RMW; Bister and 245 Emanuel 2002).

Next, we calculate the saturation convective available potential energy at the radius of maximum winds, $CAPE^*$. This calculation assumes the parcel is lifted directly from the sea surface, such that $T = T_s$ and $r = r_s(T_s)$ (where $r_s$ is found given $T_s$ via Eqs. 5 and 4). pyPI defines the outflow temperature level (OTL) and $T_0$ as the LNB and $T_{LNB}$ found during the final





iteration of $CAPE^*$ computation, respectively Note that the OTL and $T_0$ and could instead be defined from the $CAPE_{env}$
computation. Flexibility in the outflow definition is a planned improvement for pyPI. The choice to use $CAPE^*$ follows from
defining the outflow level with a fully saturated parcel lifted directly from the sea surface (see Sect. 6.2).

The ratio of sea-surface and outflow temperatures in Eq. 2 represents the scaling of PI by dissipative heating, which increases
PI when $\frac{T_s}{T_0} > 1$ (Bister and Emanuel, 1998). At each iteration this ratio is set with the fixed input $T_s$ and the current $T_0 \approx$
$T_{LNB}$. The relevance of this ratio for the PI calculation is set by the user with the adjustable parameter `diss_flag` (Sect. 4.1).
If dissipative heating is permitted to impact the tropical cyclone potential intensity (`diss_flag`=1) then the ratio remains as
defined above. If dissipative heating is not considered (`diss_flag`=0) then the algorithm assumes $T_s/T_0 \approx 1$ in the following
calculations.

Next, $p_m$ is estimated in each iteration following Eq. 3. The surface environmental virtual temperature, $T_v$, is found as the
average of virtual temperatures over the mean layer composed of the parcel (with temperature, $T_s$) and the lowest level, i.e.
$T_v = \frac{1}{2}(T_{v,s} + T_{v,j=0})$. The virtual temperature is identical to the density temperature (Eq. 10) at the surface (as $r_T = r(T_s)$),
and may be approximated at the lowest level with $r_T \approx r_{j=0}$. Combining Eqs. 2–3 to solve for $p_m$, the algorithm iterates
towards a new pressure estimate. If the number of iterations exceeds 200 (an excessive number of iterations), or if the estimated
pressure drops below an unphysical 400 hPa, then the PI algorithm fails to converge and returns missing outputs.

When the algorithm has successfully converged on a stable value of $p_m$, then the final central minimum pressure, $p_{min}$, is
set. Assuming cyclostrophic balance and that the azimuthal velocity in the eye is given by $V = V_{max}(\frac{R}{RMW})^b$, we follow a
power law scaling with exponent, $b$ (see also Emanuel 1995, their Eqs. 25–26):

$$p_{min} = p_{msl} * e^{(-\frac{CAPE|_{RMW} + \frac{1}{2}(1+\frac{1}{b})V_{max}^2}{RdT_v})}, \tag{14}$$

where pyPI assumes following Bister and Emanuel (2002) that $b = 2$.

Note that the difference, $(CAPE|_{RMW} - CAPE_{env})$, is typically small. Historically, when $CAPE_{env}$ was used to compute
PI in the final term of Eq. 2, it was found to add noise to the PI algorithm output (Kerry Emanuel, *personal communication*).
Therefore, pyPI instead replaces this term with $(CAPE^* - CAPE|_{RMW})$ in the PI computation for tractability. PI calculations
with the original Bister and Emanuel (2002) formulation have identical OTLs and outflow temperatures, but tend to have higher
$V_{max}$ values by between 0 and 32 ms$^{-1}$ (not shown). Global and tropical (20°S-20°N) mean biases from this approximation
are ~3 ms$^{-1}$ and ~2.5 ms$^{-1}$, respectively.

Finally, we may find tropical cyclone potential intensity. Assuming that the raw computed maximum gradient wind speeds
are scaled to 10 m winds with some fraction, we multiply Eq. 2 by `V_reduc` (Sect. 4.1). This step completes the PI compu-
tation. The module sets the flag to indicate the successful computation of pyPI's algorithm and then outputs $V_{max}$, $p_{min}$, the
flag, $T_0$, and the OTL.



| Inputs: | | | | |
|---|---|---|---|---|
| Symbol | Name | pyPI variable | Units | Values |
| $T_s$ | Sea-surface temperature | `SSTC` | °C | — |
| $p_{msl}$ | Mean sea-level pressure | `MSL` | hPa | — |
| $T(p)$ | Temperature profile | `T` | °C | — |
| $r(p)$ | Mixing ratio profile | `R` | g/kg | — |
| $\frac{C_k}{C_D}$ | Ratio of exchange coefficients | `CKCD=0.9` | unitless | 0.17–1.05 |
| — | Ascent process proportion | `ascent_flag=0` | fraction | 0.0–1.0 |
| — | Dissipative heating flag | `diss_flag=1` | — | 0 or 1 |
| — | Reduction of Gradient Winds | `V_reduc=0.8` | fraction | 0.0–1.0 |
| — | Upper level pressure bound | `ptop=50` | hPa | $< 100$ |
| — | Missing data flag | `miss_handle=1` | — | 0 or 1 |
| Outputs: | | | | |
| $V_{max}$ | Potential intensity | `VMAX` | ms$^{-1}$ | — |
| $p_{min}$ | Minimum central pressure | `PMIN` | hPa | — |
| — | Algorithm status flag | `IFL` | — | 0, 1, 2, or 3 |
| $T_0$ | Outflow temperature | `TO` | K | — |
| OTL | Outflow temperature level | `OTL` | hPa | — |

**Table 1.** Input/output variables and adjustable algorithm parameters for the PI module. Default parameter values are specified in the "pyPI variable" column. Parameters adjusted by the user *should never* be set outside the "Values" column prescriptions without physical justification and/or appropriate module modification.

## 4    Python Implementation

pyPI is written in Python v3.7 and its calculations are optimized with Numba (Lam et al., 2015). An average model run time is ∼13.7 seconds per 100,000 input profiles. Modeling the maximum intensity of a tropical cyclone with pyPI requires input environmental state variables: temperature ($T$) and mixing ratio ($r$) profiles on pressure levels ($p$), and concurrent $T_s$ and mean sea-level pressures ($p_{msl}$). Algorithm variables and parameters are shown in Table 1.

### 4.1    Adjustable Parameters

pyPI includes six adjustable parameters that may set in the module call, with the caveat that each should be chosen within the defined "Values" column of Table 1. Parameters set outside these values could result in syntax errors or logical errors in the output, or may give rise to unphysical PI estimates.





### 4.1.1 CKCD (default=0.9)

The ratio $\frac{C_k}{C_D}$ an uncertain constant which depends on the sea state and linearly scales potential intensity; its value is an

ongoing area of field and theoretical research (Emanuel, 2003). Table 1 includes the $1\sigma$ range of the ratio found with energy and momentum budget methods by the 2003 Coupled Boundary Layers Air–Sea Transfer (CBLAST) field program, Bell et al. (2012). Studies exploring PI variability typically use a default value of $\frac{C_k}{C_D}$=0.9 when calculating PI (e.g. Wang et al., 2014; Wing et al., 2015; Gilford et al., 2017).

### 4.1.2 ascent_flag (default=0)

The ascent process proportion determines whether the air parcels displaced in each $CAPE$ calculation (cf. Sect. 3.2) follow reversible adiabatic ascent (ascent_flag=0) or pseudoadiabatic ascent (ascent_flag=1). In the case of reversible ascent, the full moist entropy of the buoyant parcel is conserved along its displacement following a moist adiabat. In pseudoadiabatic ascent the heat capacity of liquid water is neglected. Liquid water is assumed to fall out of the parcel as it condenses, while the parcel ascends following the pseudoadiabatic moist adiabat; for more details see Emanuel (1994), their Sect. 4.7. For practical

applications of pyPI, ascent_flag may be set to any value between 0.0 and 1.0, such that the proportion of ascent is any fraction intermediate to fully reversible and fully pseudoadiabatic ascent.

### 4.1.3 diss_flag (default=1)

The dissipative heating flag determines whether dissipative heating is accounted for (diss_flag=1) or ignored (diss_flag=0) in potential intensity theory (see Bister and Emanuel 1998, their Eq. 22). When dissipative heating is included in the PI cal-

culation, the leading factor in the BE02 algorithm (Eq. 2) is $(\frac{C_k}{C_D} * \frac{T_s}{T_0})$, where $\frac{T_s}{T_0} > 1$. In the absence of dissipative heating, the leading factor is $(\frac{C_k}{C_D} * 1)$ following the original findings of Emanuel (1986, 1995). PI is often considerably lower when dissipative heating is neglected.

### 4.1.4 V_reduc (default=0.8)

Raw potential intensities are maximum gradient wind speeds (Emanuel, 2000). Therefore, gradient winds calculated with the

BE02 algorithm are not directly comparable with observed intensities at the near-surface without applying an approximate scaling between gradient and 10 meter winds. Following Powell (1980), a crude reduction of 20% (V_reduc=0.8) is typically applied to scale PI for comparison with near-surface winds. The percent reduction in the gradient wind in terms of V_reduc is $P_{reduc} = 100\% * (1 - \text{V\_reduc})$. Note that for some applications of PI, such as using it as thermodynamic parameter in climate science—e.g. incorporation into the Genesis Potential Index, Camargo et al. (2007)—V_reduc should be set to 1.0

(no reduction).





### 4.1.5 `ptop` (default=50)

The upper level pressure bound is the minimum pressure below which the input profile is ignored during PI computation. Theoretically modeled tropical cyclone outflow can often exceed the tropical tropopause on climatological timescales (Gilford et al., 2017), so setting `ptop`> 100 hPa it is not advisable. Reducing the number of considered levels by increasing `ptop` may

potentially increase the speed of calculations, at the risk of finding a too-low OTL and too-warm $T_0$. Before altering `ptop`, users should consider their particular application and the outflow levels they anticipate given the stability of their input profiles.

### 4.1.6 `miss_handle` (default=1)

The missing data flag prescribes how missing values are handled in the $CAPE$ calculation (discussed below). Following the BE02 MATLAB code (`miss_handle`=0), if missing values are found in the input temperature profile, then the algorithm will

attempt to calculate PI for all available levels above the missing values. However, the user may also conservatively choose that any missing values in the input profile will immediately set the entire PI calculation output to missing (`miss_handle`=1).

### 4.2 Handling Missing Data

Mirroring the output flag convention of the BE02 MATLAB code: `IFL`=1 when the PI algorithm successfully returns valid potential intensity ouputs, `IFL`=0 when the algorithm fails because the input data is improper for a PI calculation (e.g. if $T_s <$

$5°$C), and `IFL`=2 when the algorithm fails to converge.

  One major difference between pyPI and the BE02 MATLAB algorithm is the handling of missing data and the (related) flag provided in the output. By convention, missing input variables in pyPI are assigned Python's "Not-a-Number", NaN, to avoid errors. The BE02 MATLAB code default is that profiles may contain missing values (specifically temperatures on pressure levels), and the algorithm computes PI over the remaining valid levels.

Because missing values may sometimes be found at the surface—and the primary $CAPE$ calculation (Sect. 3.2) relies heavily on the assumption of lifting the parcel within the storm and environment from that level—errors could arise from estimating PI when ignoring near-surface buoyancy. In principle, PI should be calculated only over data points with existent sea-surface temperatures and lowest profile level temperatures. In practice, missing data may arise at the lowest profile level, which would lead to errant PI calculations if these profiles are input to the BE02 MATLAB code.

pyPI addresses this challenge in three ways. First, an adjustable parameter (`miss_handle`) is implemented to allow the user to specify how pyPI handles the missing values. If `miss_handle`=0, the code attempts to handle missing values akin to the way that the BE02 MATLAB code did, although there still remain some differences in the outputs between pyPI and the MATLAB algorithm. Specifically, pyPI's $CAPE$ calculation proceeds as normal only as long as there are no missing values between the lowest valid (non-missing) level and the OTL; otherwise, $CAPE$ module outputs (and hence PI module outputs)

are returned as missing. Second, if `miss_handle`=1, then the $CAPE$ function will automatically interpret temperature profiles with missing data as invalid, and return missing values to the PI algorithm, resulting in the PI outputs being set to missing in the return. Third, a new output flag value (`IFL`=3) is introduced in pyPI which is returned when missing values in



the temperature profile results in a missing output return from the PI module (i.e. in either of the two cases described above), which aids in data analyses.

Figure 2 shows an example of the output algorithm status flags from pyPI calculations with a month of MERRA2 data (Sect. 5.1), and with the default `miss_handle=1`. The figure illustrates the few global points which had at least some missing data, resulting in a missing PI return from pyPI (IFL=3; red grid points). In contrast, these locations have an output (but likely errant) PI from the BE02 MATLAB code. The majority of locations where missing input data results in missing output PI are near land (e.g. the Caribbean and Indo-Pacific), where missing values arise as an artifact of the differences between the sample

data and the land-sea mask applied (Sect. 5.1). Missing values in the sample data (which has lowest data pressure level of 1000 hPa) could also be in locations where the monthly average $p_{msl}$ is below 1000 hPa, resulting in $T(1000$ hPa$)$=NaN.

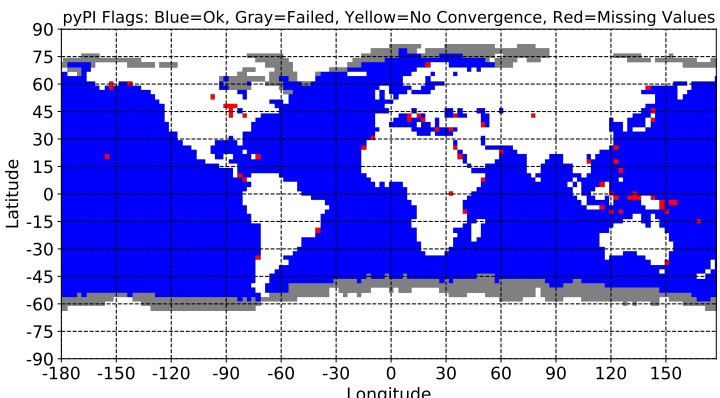

**Figure 2.** pyPI status flags from September 2004 potential intensity calculations when `miss_handle=1`. Blue grid cells indicate the PI algorithm converged, gray grid cells indicate the PI algorithm failed to pass a check, yellow grid cells indicate the PI algorithm did not converge, and red grid cells indicate the PI algorithm failed due to missing profile data.

In the example pyPI calculation (Fig. 2) there are no inputs for which the algorithm does not converge (cf. Bister and Emanuel 2002). One final complication is that BE02 MATLAB code occasionally returns $V_{max} = 0$ as an output. In these cases, pyPI instead returns $V_{max} = $ NaN.

pyPI outputs valid (non-missing) potential intensities over ~65.6% of the ocean grid points in the global 2004 sample dataset—compared with ~64.41% returned by the BE02 MATLAB code. In addition to how missing data is handled, output differences may also arise from slight variations in numerical computation between Python and MATLAB. pyPI validation tests (section 5) are computed over all spatio-temporal locations for which both algorithms have non-missing/non-zero potential intensities (and with `miss_handle=1`).





## 5 Validation

### 5.1 Sample Reanalysis Data

The pyPI sample data are monthly means of state variables from the second Modern-Era Retrospective Analysis for Research and Applications (MERRA2, Gelaro et al. 2017) in 2004, interpolated onto a 2.5° x 2.5° global grid (Gilford et al., 2017). Note that for these example pyPI calculations the water vapor mixing ratio, $r$, is approximated by substituting in the reanalysis specific humidity, $q$ (as $q \equiv \frac{r}{1+r} \approx r$, because $r \ll 1$).

Potential intensity calculations are generally linear, i.e., mean potential intensities may be estimated as a function of mean environmental variables,

$$E[V_{max}(T_s, p_{msl}, p, T, r)] \approx V_{max}(E[T_s], E[p_{msl}], E[p], E[T], E[r]), \tag{15}$$

where $E[\cdot]$ is the expected value of a function or variable. Using monthly mean environmental conditions to compute climatological monthly means of potential intensity (and the algorithm's other output variables) generates a small bias of $<1$ ms$^{-1}$ globally and $<0.5$ ms$^{-1}$ in the tropics (Gilford et al., 2019). PI's linearity property is convenient, because it reduces the scale of data needed to compute PI: daily or hourly data are not needed for monthly or longer (climatological) applications. Applications on shorter (e.g. operational or daily) timescales, however, should use appropriately shorter frequency inputs to the pyPI algorithm.

For consistency with Gilford et al. (2017), the sample data uses the land-sea mask from the European Centre for Medium-Range Weather Forecasts Interim (ERA-Interim, Dee et al. 2011) on a 2.5° x 2.5° global grid. By definition, $V_{max} \equiv 0$ where $T_s = $ NaN (i.e. over land); in some cases (e.g. if skin temperatures valid over land are used in lieu of sea-surface temperatures) PI may be mistakenly calculated over land with the PI module. In these cases, users should assign all PI algorithm outputs over land to the missing value in post-processing. As an alternative, in this pyPI example input variables over land are set to missing in a pre-processing step. Note that the mismatch between using the ERA-I land-sea mask and MERRA2 data in this example results in a set of minor output artifacts caused by missing (MERRA2 land grid points) input data over ERA-I defined ocean grid points. This artifact provides a useful demonstration of the missing data flag employed in pyPI (Sect. 4.2).

### 5.2 Validating Against the BE02 Implementation

Accompanying the environmental conditions in the sample data are outputs from the BE02 MATLAB code written by Kerry Emanuel (see ftp://texmex.mit.edu/pub/emanuel/TCMAX) and revised by Daniel Gilford for climatological research applications (see Gilford et al. 2017, 2019).

Potential intensities calculated over September 2004 with pyPI and the extensively-used BE02 MATLAB code are compared in Figure 3 over the globe; their difference is computed and plotted in Figure 4. There is excellent agreement between the two algorithms; 98.5% of output potential intensity have absolute differences $< 0.01$ ms$^{-1}$. Potential intensities calculated with the Python algorithm exhibit a slightly negative bias relative to the MATLAB calculations, but these differences are negligible compared with other uncertainties in the PI calculation, such as the ratio of surface exchange coefficients (Table 1).



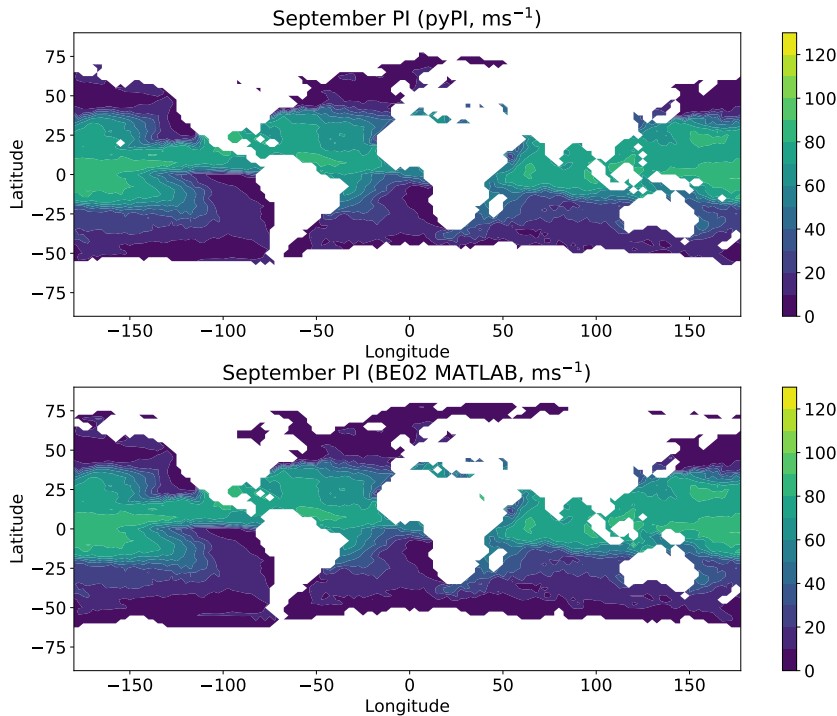

**Figure 3.** September 2004 mean potential intensities (ms$^{-1}$) calculated with pyPI (upper) and the BE02 MATLAB code (lower).

Figure 5 shows the scatter between all potential intensity values calculated with the two algorithms over the sample data, plotted against a 1-to-1 line (values lying on this line exhibit perfect agreement). The R-squared of this comparison is R$^2 \approx 1.0$ to seven significant digits, such that the calculations are nearly identical. All other output variables (cf. Table 1) from the two

400    algorithms have similarly strong levels of agreement.

A minor PI difference between pyPI and the BE02 MATLAB algorithm arises when the pyPI-found outflow level has a higher pressure and warmer temperature than found by the BE02 code. The outflow property differences result from pyPI's correction of a minor error that was present in the BE02 algorithm, where $\epsilon$ was defined as 0.622 rather than directly calculated as $\frac{R_d}{R_v}$. The small rounding error results in a handful of profiles with lower altitude outflow and lower PI values calculated by

405    pyPI. In the absence of correcting this error, the correlation between the two calculations is R$^2 \approx 1.0$ to thirteen significant digits, and the absolute maximum difference anywhere is $4.1 \times 10^{-5}$ ms$^{-1}$.

I conclude that the PI calculations made with the pyPI algorithm are adequately validated against the BE02 MATLAB code, and that pyPI is sufficiently accurate for use in research applications.




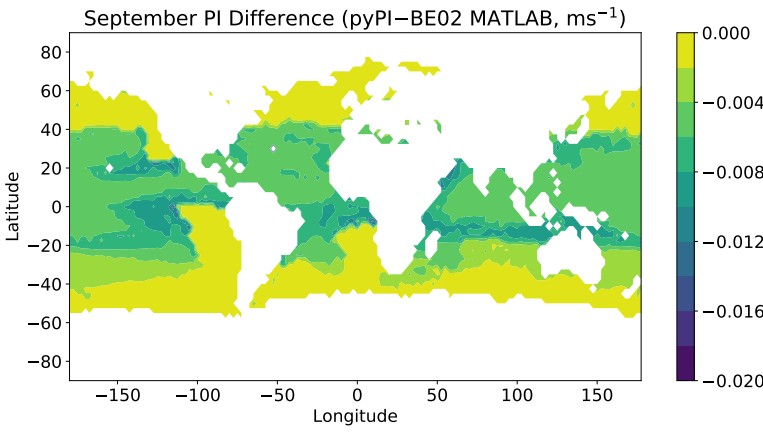

**Figure 4.** September 2004 mean potential intensity differences ($ms^{-1}$) between those calculated with pyPI minus those calculated with the BE02 MATLAB code.

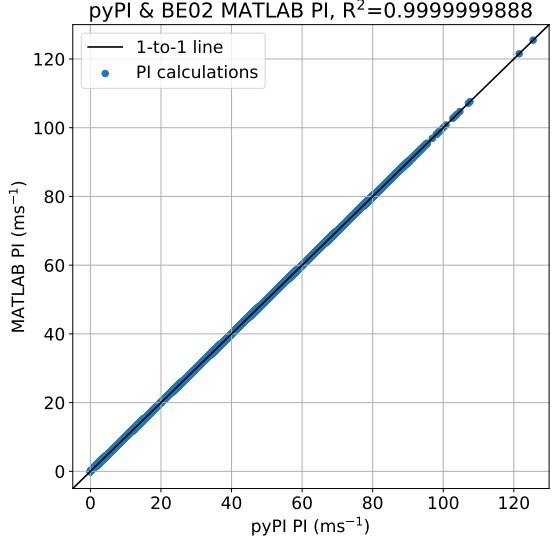

**Figure 5.** 2004 mean potential intensities ($ms^{-1}$, blue dots) calculated with pyPI (horizontal axis) and the BE02 MATLAB code (vertical axis). The black curve is the 1-to-1 line.





# 6    Example analyses

## 6.1    Annual Mean PI

2004 annual-mean sea-surface temperatures, and pyPI calculated potential intensities, outflow temperatures, and outflow tem-
perature levels are shown in Figure 6. The familiar pattern of warm SSTs in the tropics corresponds with high $V_{max}$ values,
suggesting that on an annual timescale PI is strongly influenced by $T_s$. These warm and high-PI regions are accompanied by
outflow temperature levels with annual pressures below 100 hPa, deep in the tropical tropopause region (e.g. Fueglistaler et al.
2009). Near the tropical tropopause, annual mean outflow temperatures are remarkably cold, around 200 K. On average, the
coldest outflow temperatures are found in the Western North Pacific basin, where consistent deep convection and stratospheric
circulation act to keep tropopause temperatures very cold and highly variable (e.g. Randel and Jensen 2013).

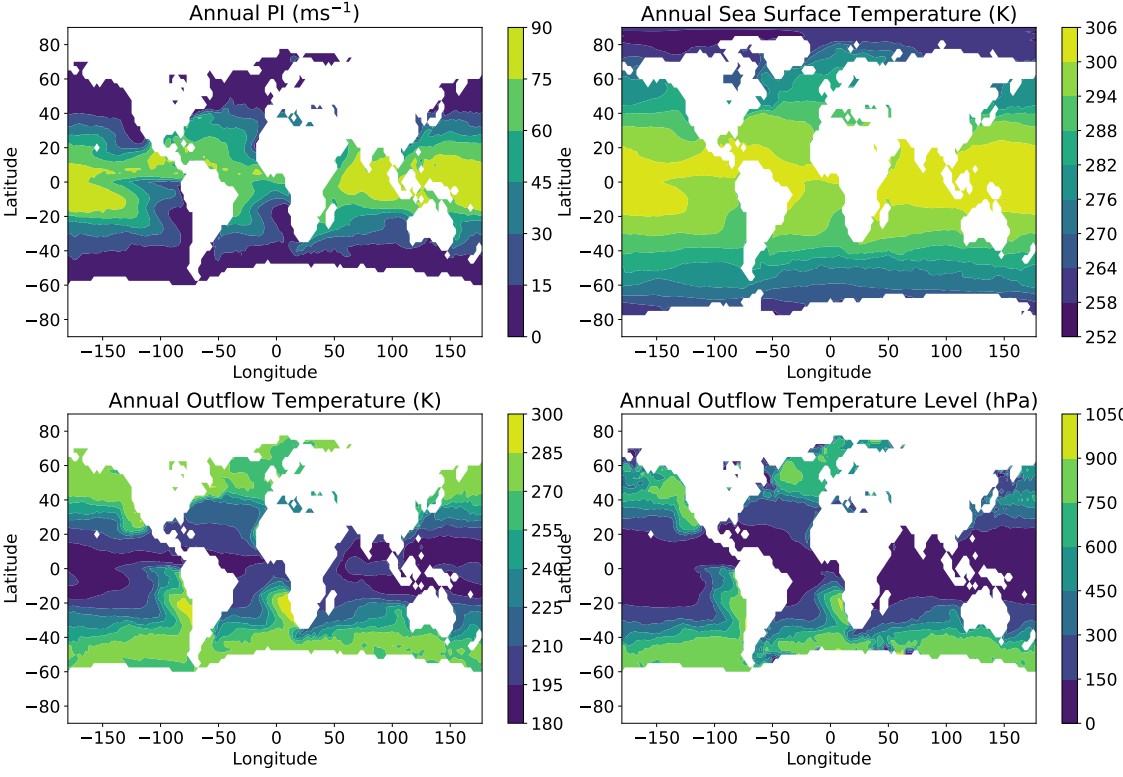

**Figure 6.** 2004 annual mean potential intensities (ms$^{-1}$, top left), sea-surface temperatures (K, top right), outflow temperatures (bottom left,
K), and outflow temperature levels (bottom right, hPa) calculated with pyPI.





## 6.2 PI Seasonal Cycles

A slightly more sophisticated application of pyPI is the calculation of potential intensity seasonal cycles. Reproducing the
methodology of Gilford et al. (2017) with pyPI calculations over 2004, Figure 7 shows the seasonal cycles of sea-surface
temperatures, and outflow temperatures, outflow temperature levels, and potential intensities in 2004 averaged over tropical
cyclone main development regions (defined in Gilford et al. 2017, their Table 1).

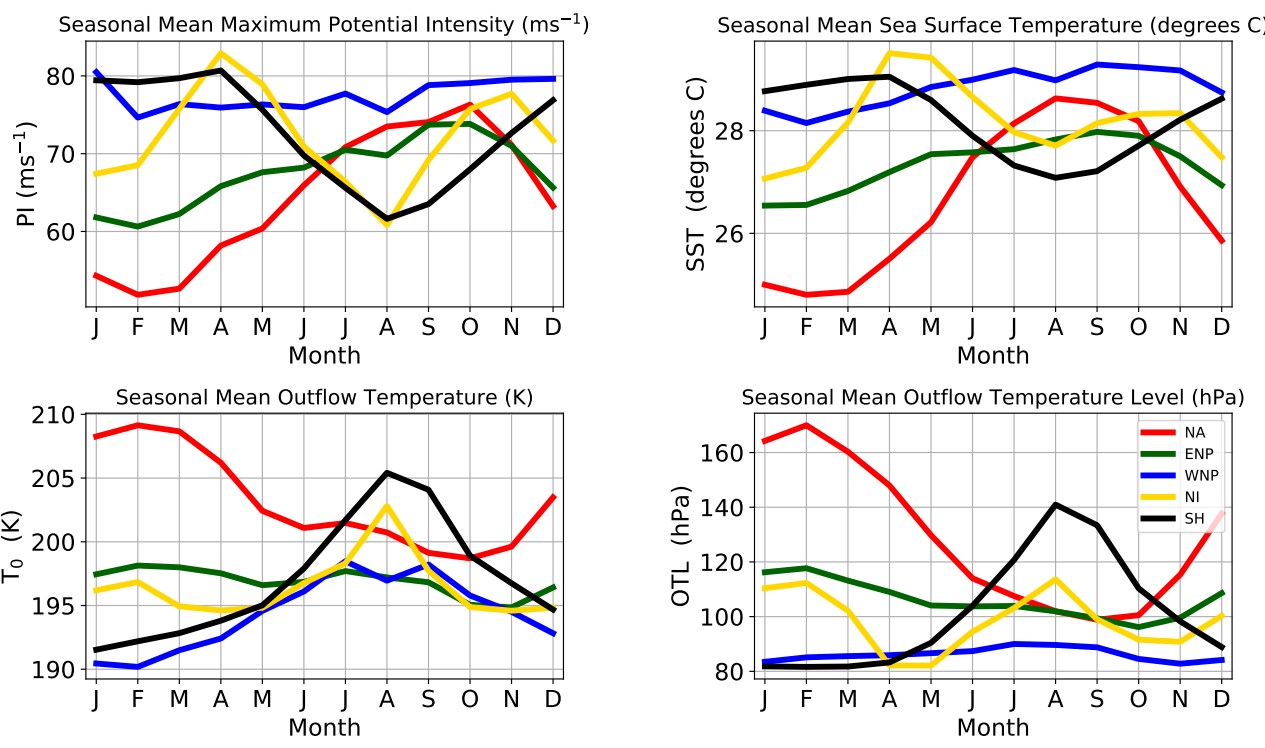

**Figure 7.** 2004 seasonal cycles of potential intensity (ms$^{-1}$, top left), sea-surface temperature (K, top right), outflow temperature (bottom left, K), and outflow temperature level (bottom right, hPa) calculated with pyPI and averaged over the main development regions defined in Gilford et al. (2017) (their Table 1): the North Atlantic (red), Eastern North Pacific (green), Western North Pacific (blue), North Indian (yellow), and Southern Hemisphere (black).

The seasonal cycles of PI are known to be quite robust year-over-year and exhibit clear differences between regions. Consistent with the findings of Gilford et al. (2017), the Western North Pacific has a nearly flat seasonal cycle of PI, while the other
basins are more intraseasonally variable. While the muted sea-surface temperatures certainly play an important role in this
damped cycle, the outflow temperature pattern is typical of the cold-point tropopause seasonal cycle (e.g. Yulaeva et al. 1994;
Randel and Wu 2014)—which the OTLs are reaching—which acts to damp the seasonal cycle further by decreasing PI in the



boreal summer and increasing PI in the boreal winter. As a result, tropical cyclones in the Western North Pacific have higher

speed limits during the boreal winter months. Consistent with this finding, historical observed typhoons show intense wind

speeds during the winter and spring months (Gilford et al., 2019). For example, in early April 2004 Typhoon Sudal reached

category 4 strength, $\sim$67 ms$^{-1}$, when the co-located monthly average PI was about 75 ms$^{-1}$.

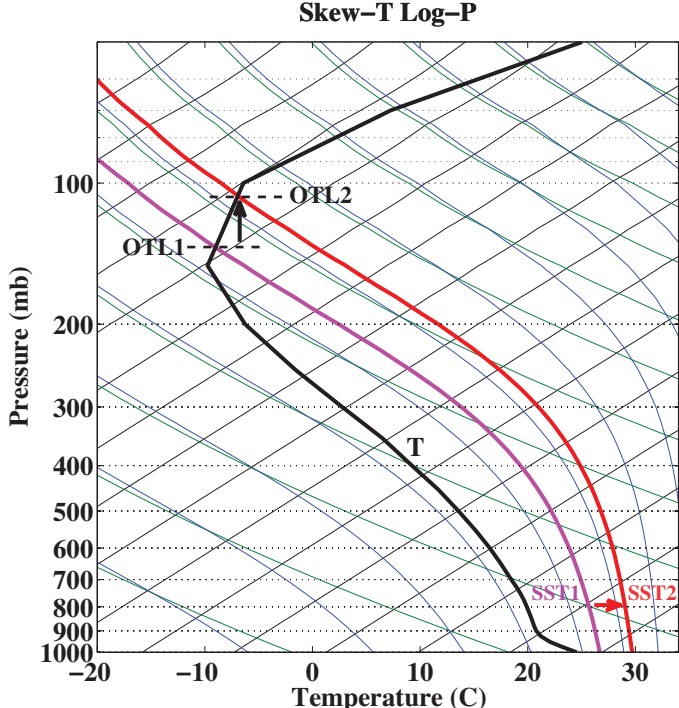

**Figure 8.** Skew-T log-P thermodynamic diagram with isotherms (thin black curves), dry adiabats (green curves), and moist adiabats (blue curves). The bold black line is a mean environmental temperature profile from the North Atlantic region (Gilford et al. (2017), their Table 1), the magenta curve is the moist adiabat associated with a mean North Atlantic sea-surface temperature (SST), and the red curve is the moist adiabat associated with a sea-surface temperature 3°C warmer than the mean.

The seasonal cycles of each basin illustrate the complex relationship between sea-surface temperatures, OTLs, and outflow

temperatures. Figure 8 diagrams this relationship in more detail, showing how one assumption in the pyPI algorithm impacts

the output PI values. pyPI assumes that the outflow temperature and its level are derived by finding the LNB assuming a

saturated parcel lifted from the sea-surface with temperature, $T_s$. This implies that, following a moist adiabat, the level of the

neutral buoyancy is a function of only $T_s$ and the environmental temperature profile, $T$. Given a fixed temperature profile, then

an increase in $T_s$ (e.g. from SST1 → SST2 in Fig. 8) requires that the associated OTL will be found at a higher altitude (OTL1

→ OTL2 in Fig. 8) and the associated outflow temperature will likewise change. As the atmosphere's stratification increases

into the lower stratosphere, increases in $T_s$ become less effective at changing the OTL and its temperature, with the effect nearly

saturating when the outflow reaches the cold-point tropopause (e.g. 100 hPa in Fig. 8). At this point, $T_0$ variability is almost





completely decoupled from $T_s$ variability. Instead these $T_0$ values become influenced by tropopause region variability (e.g. Emanuel et al. 2013; Ramsay 2013; Wang et al. 2014; Wing et al. 2015; Gilford et al. 2017) which is controlled by radiation, dynamics, and deep convection (e.g. Fueglistaler et al. 2009; Randel and Wu 2014).

These properties are borne out in the example 2004 seasonal cycles computed in Fig. 7. In the North Atlantic basin sea-
surface temperature and OTL seasonal cycles are inversely proportional: colder sea-surface temperatures have higher pressure OTLs and warmer outflow temperatures found in the upper troposphere (where $\frac{dT}{dz} < 0$) in all months except August-September. In these late summer months the OTL reaches near the cold-point tropopause, and the outflow temperature slightly increases, following the seasonal cycle of warmer tropopause temperatures (e.g. Yulaeva et al. 1994). A contrasting pattern is observed in the Western North Pacific, where OTLs have almost no seasonal cycle: in this basin the calculated outflow *always*
reaches the lowermost stratosphere (OTL $\leq$ 90 hPa). Accordingly, the outflow temperature seasonal cycle perennially follows the seasonality of lowermost stratospheric temperatures, which minimize in the boreal winter and maximize in the boreal summer (Yulaeva et al., 1994). Comparing with Eq. 1, this $T_0$ seasonality leads to relatively *increased* PI values in the boreal winter and relatively *decreased* PI values in the boreal summer. Overall, the PI seasonal cycle in the Western North Pacific is damped over the year, a pattern that is observed in real-world tropical cyclone intensities (cf. Gilford et al. 2019).

**6.3 Decomposition Analysis**

The relative contributions to potential intensity may be mathematically derived by decomposing Eq. 1. Taking the natural logarithm of both sides:

$$2 \times log(V_{max}) = log(\frac{C_k}{C_D}) + log(\frac{T_s - T_0}{T_0}) + log(h_o^* - h^*), \tag{16}$$

then PI variability is related to variability in either tropical cyclone efficiency ($\frac{T_s - T_0}{T_0}$) or thermodynamic disequilibrium ($h_o^* -$
$h^*$); recall that $\frac{C_k}{C_D}$ is taken as a constant. As an example following Gilford et al. (2017), Eq. 16 is applied to pyPI calculated 2004 seasonal cycles of potential intensity (from Fig. 7). pyPI calculates PI directly, and efficiency may be directly computed from input $T_s$ and output $T_0$; following Wing et al. (2015) the disequilibrium term is taken as a residual from Eq. 16.

After finding each term in Eq. 16 over each basin and seasonal cycle, the amplitude (defined as the annual range evaluated with monthly observations) of each seasonal cycle is plotted in Figure 9. By convention (Gilford et al., 2017, 2019), a negative
amplitude indicates the ~sinusoidal seasonal cycle reached its maximum in the boreal winter and minimum in the boreal summer.

In all basins, the disequilibrium term drives the largest portion of the seasonal amplitude. This is consistent with sea-surface temperature seasonal cycles which dominate the disequilibrium variance (Fig. 7). The efficiency term is smaller, and in each basin it follows the same cycle as thermodynamic disequilibrium, with the exception of the Western North Pacific (where the
efficiency seasonal cycle maximizes in the boreal winter and minimizes in the boreal summer). This opposite signed seasonality between disequilibrium and efficiency in the Western North Pacific is directly related to the influential seasonality of the near-tropopause outflow temperatures found with the pyPI calculations (Sect. 6.2, see full discussions in Gilford et al. 2017,



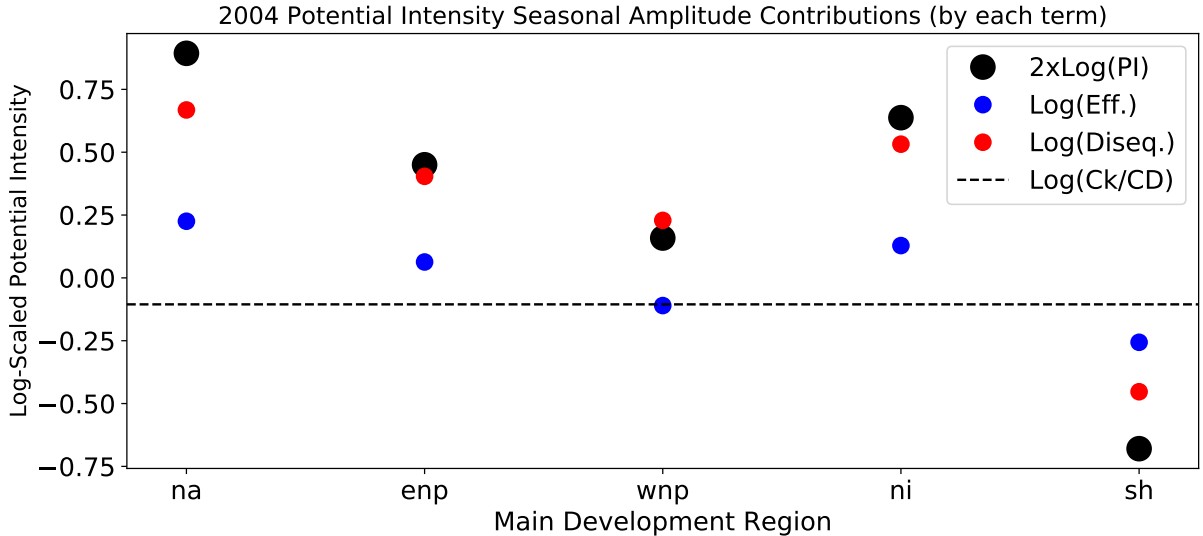

**Figure 9.** Seasonal amplitudes of each PI decomposition term (Eq. 16) in 2004 and each main development region, calculated with pyPI. Compare with Gilford et al. (2017), their Table 2. By convention, negative amplitudes indicate the associated seasonal cycle peaks in the boreal winter. For reference, the dashed black line indicates the magnitude and sign of the seasonally invariant $\log(\frac{C_k}{C_D})$.

2019). Notably, the Southern Hemisphere shares this outflow temperature seasonality, which actually amplifies the efficiency seasonal cycle through both sea-surface temperatures and outflow temperature intraseasonal variability. In all other basins,

outflow temperature seasonality is offset by the sea-surface temperature seasonality, which acts to mute the efficiency term and further contribute to disequilibrium dominating their seasonal cycles. The decomposition in Fig. 9 illustrates how the roles of environmental conditions in PI seasonality are basin dependant.

These simple examples show how pyPI may be used to study tropical cyclone intensities, and likewise demonstrate pyPI's ability to produce findings similar to those previously computed with the BE02 MATLAB code.

# 7    Summary, Limitations, & Future Development

pyPI is a Python package that models the maximum potential intensity (PI) of a tropical cyclone given its environmental conditions. pyPI(v1.3) is the first fully documented PI algorithm, advancing on the previous MATLAB code (Bister and Emanuel, 2002) which has been extensively used but under-documented in the literature. In addition to documenting PI computation, allowing dynamic parameter selection, and correcting minor errors in the previous algorithm, pyPI is also an open source,

maintained, and archived project which permits reproducibility, continual updates and improvements, and accountability for future PI calculations in the tropical meteorology community (Gilford, 2020).





pyPI calculations exactly reproduce outputs from the Bister and Emanuel (2002) algorithm, except in rare cases where the original algorithm's implementation was errant. Sample analyses show the flexibility and usefulness of PI calculations for understanding variability and thermodynamic contributions to climatological tropical cyclone maximum intensities.

Because of its statistical ties with observed storms (Emanuel, 2000; Wing et al., 2007; Gilford et al., 2019; Shields et al., 2019) PI is powerful tool for exploring past and future changes in real-world maximum intensities. pyPI computations have a broad range of possible applications, which could include operational meteorology (e.g. the PI maps produced by the Center for Land–Atmosphere Prediction, http://wxmaps.org/pix/hurpot, Emanuel et al. 2004) and climate change research (Sobel et al., 2016).

Potential intensity is a theoretical model with several notable limitations. Real-world tropical cyclones rarely are in quasi-steady state or meet the idealized conditions required for the Carnot cycle model. This makes PI less suitable for operational purposes, though it may still be incorporated into real-time genesis or intensification indices (see below). Furthermore, PI theory does not directly account for complicating factors such as vertical wind shear or large-scale subsidence, which are known to have important influences on tropical cyclone intensity. The ratio of surface exchange coefficients, $\frac{C_k}{C_D}$, is also highly

uncertain but important for PI magnitude. Previous studies have adapted PI to make it more suitable for various applications (e.g Lin et al., 2013; Kieu and Zhang, 2018); pyPI users should carefully consider PI assumptions and applicability in their research problems (Gilford, 2018), adapting pyPI or suggesting package enhancements as appropriate.

Future planned improvements of pyPI include an expansion of the codebase to compute other tropical cyclone thermodynamic and statistical indices, including the Genesis Potential Index (Camargo et al., 2007; Zhang et al., 2016) and Ventilation

Index (Tang and Emanuel, 2012). A direct disequilibrium calculation (e.g. Wing et al., 2015) module would permit comparisons with the residual approach currently employed in pyPI (Sect. 6.3). Finally, further improvements in the algorithm's handling of missing data are warranted to reduce the algorithm run time and improve pyPI's applicability for a wider range of input profiles.

*Code and data availability.* pyPI version 1.3 (Gilford, 2020) and accompanying data for validation and sample analyses are available at
https://github.com/dgilford/pyPI/releases/tag/v1.3 and archived at https://doi.org/10.5281/zenodo.3985975. pyPI is provided under the MIT license.

**Appendix A: pyPI Constants**

Constants used in to model potential intensity in the BE02 algorithm have been directly used in pyPI and are recorded in Table A1.

*Author contributions.* DG conceived the study, wrote and tested pyPI software and its v1.3 modules, notebooks, and scripts in Python, and wrote the documentation and manuscript.



| Symbol | Constant Name | pyPI variable | Value/Units |
|--------|---------------|---------------|-------------|
| $c_{pd}$ | Specific heat of dry air | CPD | 1005.7 J/kg.K |
| $c_{pv}$ | Specific heat of water vapor | CPV | 1870 J/kg.K |
| $c_\ell$ | Specific heat of liquid water | CL | 2500 J/kg.K |
| $R_v$ | Gas constant of water vapor | RV | 461.5 J/kg.K |
| $R_d$ | Gas constant of dry air | RD | 287.04 J/kg.K |
| $\epsilon$ | Ratio of gas constants | EPS | $\frac{R_d}{R_v} = 0.6219...$ |
| $L_{v0}$ | Latent heat of vaporization at $0^\circ$C | ALV0 | 2.501e6 J/kg |
| — | Pressure upper boundary | ptop | 50 hPa |

**Table A1.** Meteorological constants used in the potential intensity algorithm.

*Competing interests.* The author declares no competing interests.

*Disclaimer.* The code is made publicly available without any warranty, and permissions are provided pursuant the MIT License (https://opensource.org/licenses/MIT).

*Acknowledgements.* I am particularly grateful to Kerry Emanuel for his encouragement and permission to develop, document, and distribute pyPI, and for his development of PI theory and the original PI algorithm. DG is supported by NSF Grant ICER-1663807 and NASA Grant 80NSSC17K0698. Thanks to Daniel Rothenberg for implementing a Numba optimization (Lam et al., 2015) of the primary pyPI module. Thanks to Allison Wing, Dan Chavas, Jonathan Lin, and Raphael Rousseau-Rizzi for helpful comments and suggestions on pyPI and its documentation. This study is validated against a portion of previous research on PI seasonality, published in Gilford et al. (2017) and Gilford
et al. (2019) by the American Meteorological Society. Those works (and pyPI which followed) benefited immensely from helpful discussions with Susan Solomon and Paul O'Gorman.



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
