# Peer review of "pyPI (v1.3): Tropical Cyclone Potential Intensity Calculations in Python"

_Geoscientific Model Development, 2020_

## Referee Comment (RC1) · Anonymous Referee #1 · 23 Nov 2020

Summary: this work presents a development of the theoretical maximum potential intensity (PI) code, which is written and maintained on Github by the author. All aspects of the PI calcuations along with discussions are provided in great details, which present the author's significant effort in explaining and implementing the PI code that is worth consideration. This work provides the TC research community a good tool for studying the climatology of TC intensity, and so I would recommend it for publication on GMD. I have only a few comments that the author may want to take them into account so this PI package can be more useful for future research and applications.

Major comments:

1. The author may consider including a list of the functions that correspond to each calculation step listed in Table 1 so readers can quickly locate the part where they

want to make a change. This will be very helpful for the research community, because most often people want to implement or examine different processes rather than simply running it as a black box. So, being able to modify the code easily is important and useful in the long run for users.

2. In the abstract, the author mentioned about a future plan to improve pyPI's flexibility, which I think is really beneficial for users to examine the validity of different assumptions in Emanuel's PI framework (see also line 125-126). In this regard, it would be great if the author can document in the code where those assumptions in Emanuel's PI framework can be modified so users can examine these assumption in details. For example, a recent work by Kieu and Wang (2017, JAS) showed that the moist neutrality may not be applied, even at the mature stage. The inclusion of this factor can be done very easily within the PI code by introducing a factor $(1 - \alpha * \Gamma)$ in the Vmax expression, where $\Gamma$ is the environmental lapse rate and the factor $\alpha$ can be treated as an empirical parameter. The author can therefore extend his code to allow for such a flexibility so readers can explore the tropospheric stratification, as including any such environmental variable in the PI would introduce a new climate variable to the PI code and increase the value of the pyPI code. Please note that this comment is more or less a suggestion to improve the capability of this PyPI code, and by no mean critical. So, it is up to the author to provide a support for any way that can help verify the PI assumptions.

Minor comments

In Eq .(14), there is a factor $RdT_v$ in the denominator of the exponent. What is the parameter $d$ in the denominator here?
* * *

---

## Referee Comment (RC2) · Anonymous Referee #2 · 22 Dec 2020

This paper documents PyPI, a Python port/update of a commonly used set of Fortran and Matlab utilities maintained by Kerry Emmanuel's group. The author first discussed the algorithmic implementation of the PI calculations and then highlights a couple brief examples/applications of the code using reanalysis data.

While the code or PI theory certainly isn't new, the author notes that the implementation of the algorithm itself has historically been relatively opaque. I particularly appreciated the noting of simplifications/approximations (e.g., the hard-coded Rd/Rv ratio corrected, the LCL estimation, etc.), which are generally buried in code and rarely discussed in scientific publication.

I will note that the paper itself reads a bit more like a software documentation or user guide than a traditional scientific paper. Personally, I feel this is acceptable for a journal

such as GMD and find the effort to make software open source laudable. Traceability is going to grow more important over coming years, particularly as data volumes get large enough that archiving post-processed data is not feasible (i.e., the version of software used to process data can be tagged, such that a reader can download the raw source data plus software and recreate published findings). Some reviewers may have other opinions on the utility of such publications.

In general, I find the paper thorough, clean, and relatively pleasant to read. I feel it is acceptable for publication in GMD after some minor revisions, generally focused on some clarification of some points raised by the manuscript.

Minor comments:

Line 40 (etc.). I am not sure directly linking web addresses (see also 47-48) is useful in-line. I routinely stumble upon dead links in papers published even a few years ago. My suggestion would be archiving a version of the Matlab/FORTRAN in the PyPI repository for posterity if the authors (i.e., Kerry Emanuel) are OK with it. Otherwise, web links should probably be moved to footnotes.

Line 68. Semantics, but I might refer to this as cyclones resulting *indirectly* in response to the energy imbalance. It is deep convective overturning that is the primary manifestation; TCs form in such environments (e.g., you can still simulate reasonable heat transport without explicitly simulating TCs in low-resolution models).

Line 147. There are numerous algorithms for computing CAPE in use. Many models (e.g., WRF, MPAS) have in-line diagnostic calculations (such that CAPE can be directly output from the model) while other packages (e.g., MetPy, SHARPpy, NCL) have routines for calculating CAPE offline (from profiles of T, q, etc.) as well. Does the specific choice of CAPE calculation matter? Does the method here differ from others? If one were to "plug in" the CAPE calculation from SHARPpy, would the results change?

Line 241. While it almost certainly doesn't matter in practice, 0.5 hPa (half a millibar)

strikes me as a bit large for a convergence tolerance. What is the increase in iterative cost in using a 0.05 hPa tolerance? Do the answers change at all?

Lines 281. Any idea how this timing compares to the Fortran or Matlab code? I assume the timing is quite poor for Matlab given its generally high level of abstraction, which might make PyPI look even better.

Line 289. In addition to the CBLAST reference, George Bryan wrote a very nice (and thorough) paper regarding this ratio that would be worth directing readers towards for more information (doi:10.1175/MWR-D-11-00231.1).

The LaTeX equations need a bit of work. This is probably something that can be handled in the proof stage, but Eqs. 8 and 14 have vertically compressed exponents. For functions like exp and log, the convention is to use the \exp{...} notation, which removes the italics. Eqns. 5, 9, and 10 should use \left( and \right) to vertically extend the parentheses. Eqn. 14 is difficult to read.

Line 306-307. Could include an estimate of the underestimation here. If dissipative heating is excluded the PI is about 70% of the "full" value?

Line 350. Based on later text, I assume this is a monthly average for MERRA2, but might be helpful to mention that here.

Line 376. Compared to 6-hourly? Daily?

Temperature units are inconsistent. I understand this is probably because inputs are in degC to be more "observational" but having degC as inputs and then outflow temperature be an output in K feels risky. Are there error checks in PyPI to ensure unit correctness? Luckily temperature degC/K is a pretty easy one to handle in the troposphere.

Typographical errors and grammar:

This is a commentary on style, but I found that the first-person pronouns coupled to

frequent references to previous papers by the author make the manuscript read a bit more like a user guide. Sentences like "I developed (PyPI...) to meet these needs." are an example. This is really up to editorial discretion, but (while I am typically a proponent of active voice) I think writing the paper in a passive voice might read a bit more traditionally like a peer-reviewed manuscript given the subject material.

Line 21. Earth should be capitalized.

Line 39. "Kerry" is colloquial, would replace with either full name or "the author."

Line 50. Need for *a* transparent...

Line 465. Instead of tilde, using "approximately"?

Line 483. ... used, but under-documented, in the

Eq. 14. Subscript "d"

Fig. 6. "Widen" longitudinal extent of panels to take advantage of whitespace.

The Zenodo link for the software is referenced in-line quite a bit. It is probably redundant since the point of the paper is to document the code at the Zenodo link (so referencing itself is circular). I would remove these references and just keep the DOI in the acknowledgments/data availability.

---

## Author Comment (AC1) · 5 Feb 2021

We thank the reviewer for their support for this effort and their helpful comments which have improved this work.

In response we have endeavored to clarify some opaque details of certain points which were raised by the reviewer, including the algorithm sensitivity to the minimum pressure convergence threshold, the role of dissipative heating, model run times compared with MATLAB, and the differences with other CAPE algorithms. We have also updated the equation formats, added a temperature unit check, and fixed the tone of the piece from first person to a more passive voice.

Minor comments

[Figure]

Line 40: As suggested, the MATLAB function has been included directly in the pyPI v1.3 code archive. We have removed this link and original line 40 to avoid confusion.

Line 68: Agreed. This line has been changed to, "Tropical cyclones arise as an indirect response to. . ."

Line 147: As highlighted by the SHARPpy introductory BAMS article (Blumberg et al. 2017), different thermodynamic profile analysis routines—and especially CAPE calculations—can vary substantially from one another. In our case, we expect this will also be true. While similar mathematically to other existing CAPE routines (e.g., SHARPY and MetPy), there are several key differences between pyPI's CAPE algorithm and others.

First, finding the lifting condensation level is a part of the pyPI CAPE routine, found empirically as described in Eqn. 8. This method of determining the LCL contrasts with other programs. SHARPpy, for instance, begins by finding the LCL temperature with an empirical formula (this is possibly from Hart and Korotky 1991), and then determines the p_LCL from that. MetPy has an iterative procedure which convergences on the LCL pressure as a the parcel is lifted (consistent with its temperature and water vapor properties). Across the sample dataset we find that the LCL differences between pyPI and MetPy (when the parcel is lifted from the same sfc. level$\sim$1000hPa) can be up to 8 hPa. As a result, the associated CAPE calculation will be different, as the point where the ascent goes from dry to moist adiabatic can affect the amount of integrated temperature difference.

More importantly are the specific assumptions made when computing CAPE itself. MetPy calculates CAPE assuming it is proportional to the vertically integrated temperature differences. SHARPpy assumes that CAPE is proportional to the vertically integrated virtual temperature differences. pyPI assumes that CAPE is proportional to the vertically integrated density temperature differences. In the lower part of the atmosphere the density temperature and virtual temperature will be the same (as $r\_T \rightarrow r$)

and we would expect SHARPpy and pyPI to yield similar results if they are making the same ascent assumptions (below). However, without the virtual temperature correction, the errors in the MetPy CAPE calculation could be tens of percent (Doswell and Rasmussen 1994).

pyPI also has the capability of scaling between reversible and psuedoadiabatic ascent, whereas MetPy and SHARPpy (to our knowledge) use exclusively pseudoadiabatic ascent. As noted in the manuscript's main text, this will impact the buoyancy, and make the reversible calculations in pyPI (which are typically the default in PI studies) less comparable with other packages which only look at psuedoadiabatic ascent.

We stress that it is the CAPE difference between the saturated CAPE in the eyewall and the environmental CAPE which drives the PI calculation (see also Garner 2015). We suspect that PI calculations will overall be sensitive to the definition/module of CAPE used. More work is needed to determine the sensitivity of pyPI to thermodynamic assumptions and functional forms (see response to other reviewer, where we discuss this in more detail, and following their suggestion provide a list of assumptions in Appendix B). It is our hope that the availability of the pyPI project will enable these and related studies, but a full analysis of this sensitivity is beyond the scope of this paper.

We have now added a short discussion in the beginning of the CAPE algorithm description noting that other CAPE algorithms exist (citing MetPy and SHARPpy in particular) and have differences based on particular assumptions. The distinctions in pyPI's definition of CAPE (especially the reversible ascent option and LCL definition) are also highlighted within section 3.2.

Line 241: To test the sensitivity of the calculation to this convergence threshold, we reduce its value to 0.05 hPa, as suggested, and re-run the sample analyses. Because the threshold is just an upper bound on the uncertainty of the final minimum pressure (i.e. the final $p\_m$ in Eqn. 3), we expect that the related sensitivity on V_max will likewise be small and fractional.

[Figure]

Across the full sample dataset we find that, between the calculations with convergence thresholds of 0.5 hPa and 0.05 hPa, the maximum difference in potential intensity (anywhere) is 0.26 m/s. Simultaneously, the computational cost of this tighter convergence threshold is an increase of ~3 seconds per 100k profiles on our machine (a ~29% increase in elapsed run-time). Because the PI numerical errors are much smaller than those associated with the uncertainties in other parameters/assumptions (e.g. the $C_k/C_D$ ratio), and by contrast the associated run-time costs are relatively high, we ultimately do not implement this change in pyPI v1.3. Depending on a user's preference, needs, and available computational resources, they may want to make this trade-off between code efficiency and precision in their individual pyPI computations.

Line 281: This is an excellent question, as it useful to place the pyPI speed in context.

After updating the code with small improvements and new version dependencies, the mean elapses run time (i.e. the wall time for the code's execution) is now about 10.13 seconds on our machine (a laptop).

To find the wall time associated with each pyPI, we resample (with replacement) environmental profiles from the MERRA2 2004 dataset and then run them through the pyPI algorithm. This process is repeated 10 times to assess the variance of the run-time. As we scale the number of samples computed through the algorithm, the run time appears to linearly scale; see attached Fig. R1.

The results are somewhat noisy, possibly because of system processes or sampling; ultimately 100,000 profiles takes about 10.1 seconds to calculate PI with pyPI.

We perform the same analyses with the original Bister and Emanuel (2002) PI algorithm—which is also archived within the pyPI repository, as noted above—see attached Fig. R2.

We find that the MATLAB algorithm appears less noisy, but likewise scales linearly with the number of runs processed through the algorithm. The MATLAB algorithm

takes about 8.3 seconds to calculate PI for 100,000 profiles (about 18% less elapsed run-time than for pyPI). Note that although the MATLAB algorithm outperforms pyPI, it retains the original errors in constants, and poorly handles missing data (as discussed in the main text, section 4.2). In contrast, the conditional statements which properly handle missing data in the pyPI algorithm increases its run time.

We stress that run times will ultimately depend each user's particular implementation and system. Whereas these tests provide context for pyPI's performance relative to MATLAB for our application and system, they are not necessarily representative for each application or user.

We have added a brief discussion on the mean pyPI run time (and comparisons with the MATLAB algorithm) to the main text.

Line 289: Thank you for noting this. We have added Bryan et al. (2012) and Green and Zhang (2014) as additional references for the sensitivity of numerical simulations to the $C_k/C_D$ ratio.

LaTeX Equations: Thank you for the suggested equation improvements. We have modified the formatting for these as follows:

Eq. 8—An exponential superscript (ˆ) has been added and the exponential is no longer vertically compressed

Eqs. 3, 7, 12-13, 16, and throughout text—"log" has been replaced with \log

Eqs. 5, 8-10, and 14—now have parentheses which vertically extend

Eqs. 5 and 14—"eˆ" has been replaced with \exp as suggested. This also distinguishes the exponential from water vapor pressure introduced in Eq. 4

Lines 306-307: We agree this original statement was not specific enough. As an exercise, one can determine a rough estimate of how much dissipative heating influences the potential intensity calculation. As noted in Bister and Emanuel (1998), including

dissipative heating acts to scale the enthalpy/drag flux ratio ($C_k/C_D$) by a factor of $T_s/T_0$ (see Eqn. 2). A rough scaling for this factor during a tropical cyclone season is $T_s/T_0 \sim 300K/200K \sim 3/2$. As $V_{max}$ is proportional to the square root of this quantity, we estimate that dissipative heating scales $V_{max}$ by $sqrt(3/2) \sim 1.22$, or about 22%.

We can also explore the influence of dissipative heating empirically. Using the sample input dataset in this study (from MERRA2 in 2004), we recalculate the potential intensity after turning dissipative heating off (diss_flag=0). Plotting the scatter between the valid PI calculations with and without dissipative heating against one another, we find the result in attached Fig. R3.

There is a strong approximate-linear relationship between PI calculations with and without dissipative heating. The slope of the linear fit over the full range of valid PI values is $\sim 1.34$, which is slightly larger than the rough scaling estimate above.

Looking more closely, it appears that this relationship is not entirely linear. To look at how it changes as a function of the PI calculation, we bin each profile's PI calculations with and without dissipative heating (at each individual time and grid location) by its value of non-dissipative heating PI (every 10 m/s). Then we recalculate the slope of the linear fit between the two potential intensity samples in that bin, and determine the percent difference between them (attached Fig. R4).

The slopes of these linear fits appear to be noisy, which likely comes from the changing number of profiles/samples in each bin, and the individual SST/outflow temperature properties dominating each bin. The percent difference between these potential intensities, however, is consistently around $\sim 25\%$ at intensities of tropical storm wind speeds or greater. While a full analysis of this topic is beyond the scope of this model development study, it is clear that the effects of dissipative heating are not always static—a result which is consistent with how sea conditions and wind speeds may affect the enthalpy and drag fluxes (e.g. Bao et al. 2011). It also illustrates the usefulness of pyPI for investigating these PI properties with gridded datasets.

Taken altogether, we conclude that (on average) dissipative heating will increase PI calculations by about 20-30%. Old lines 306-307 have been rewritten as: "Scaling arguments and empirical estimates suggest that dissipative heating increases PI by about 20-30% (not shown)." The above results been included in the pyPI Git repository in a Jupyter notebook (dissipative_heating_effect.ipynb).

Line 350: This illustrative analysis shows the flags for a single month (September 2004) of pyPI output. This was previously noted in the Figure 2 caption, but now we also clarify in text.

Line 376: Thank you for noting this needed specificity. We now note this is "instead of 6-hourly calculations" in the main text.

Temperature Units: The distinction of input Celsius units (for the temperature profile/SST) and output Kelvin units (for the outflow temperature) is a hold-over from the original Bister and Emanuel (2002) algorithm. As you note, we suspect the roots of this difference are that some observational input temperatures are provided in degrees Celsius (e.g. soundings). However, other sources (such as the reanalyses here) provide their raw gridded data in kelvin. In light of this, we agree that a unit check for temperatures is warranted.

To follow your suggestion, we have added a very simple set of conditional statements to the main PI algorithm which check whether the SST and the air temperature in the input profile are in degrees Celsius. They examine the input temperatures, and if any of these exceed 100 (which would be indicative of unrealistic Celsius inputs, pointing to a potential input in kelvin), then the code fails and returns missing values and the associated flag.

A long-term goal for improving pyPI (compared with this rudimentary test) is to integrate full unit support, through the use of a tool such as Pint (https://pint.readthedocs.io/en/stable/). This, however, would require an overhaul of the existing code and could also noticeably affect the run time; we therefore leave it for

future work.

Typographical and Grammatical errors

Thank you for noting the paper's original tone took away from the study. While this study is a model description paper, and hence documenting the model development process is important, we understand and agree that using these personal pronouns detracts from the overall presentation of this study.

Accordingly, throughout the study personal pronouns have been removed and those sentences have either been rewritten or "I" has been replaced with a more passive voice. We have also revised the manuscript to remove any unnecessary (too-frequent) references to our own work. In line with your comment, we hope this revision makes the paper more readable and professional, consistent with the tone of more traditional peer-reviewed manuscripts.

Line 21: Earth is now capitalized.

Line 39: Thank you for noting this was incomplete; Kerry Emanuel's full name has been added.

Line 50: Added "a"

Line 465: "approximately" now replaces the original tilde.

Line 483: These commas have been added.

Eq. 14: Subscript "_d" has been added; the equation has also been updated for readability (as noted above).

Fig. 6: While the longitudinal extent of the figure is already global we agree that we need to make the best use of the whitespace around it. We have therefore increased the overall size of the image and removed the original whitespace around the image (especially on the long/longitudinal edge).

Zenodo link: The in text reference to the Zenodo archive has been removed from the main text, and is now only included in the Code Availability section. The line (original line 123-124) discussing comments within the code, which referenced this, has been removed for readability (consistent with the tonal changes discussed above).

References: Bao, J.-W., Fairall, C. W., Michelson, S. A., & Bianco, L. (2011). Parameterizations of Sea-Spray Impact on the Air–Sea Momentum and Heat Fluxes, Monthly Weather Review, 139(12), 3781-3797.

Hart, J.A., and W. Korotky, 1991: The SHARP workstation v1.50 users guide. National Weather Service, NOAA, US. Dept. of Commerce, 30 pp. [Available from NWS Eastern Region Headquarters, 630 Johnson Ave., Bohemia, NY 11716.]

——————————————————

[Figure]

**Fig. 1.**

[Figure]

**Fig. 2.**

[Figure]

**Fig. 3.**

[Figure]

**Fig. 4.**

[Figure]

---

## Author Comment (AC2) · 5 Feb 2021

We thank the reviewer for their thoughtful comments and suggestions to improve this study, along with their positive recommendation of this work. We have sought to address the comments below, and hope the reviewer will find them to be an improvement on the previous iteration of this study, making the work more flexible and useful for the scientific community.

In particular, we have added a short appendix to note which functions are used and included in the pyPI repository, and relate them to the steps in the algorithm boxes. We have also added a short written section suggesting a few assumptions that one could modify to alter pyPI for their own tropical cyclone studies.

Major comments

1. We agree that a list of functions (and especially connecting them with the algorithm steps) would be a useful addition to the study for interested readers. We assume that the reviewer was referring to the steps in Algorithm 1 (in lieu of Table 1, which includes only the input/output variables).

In order to not detract from the flow of the main study itself—and to avoid moving the study towards a user's guide, as cautioned by the other referee—we have added this information in a short Appendix (B). We include the list of functions pyPI employs which are commonly used in meteorology: the empirical Clausius-Clapeyron equation (i.e. the Bolton eqn. for saturation vapor pressure), the latent heat of vaporization, vapor pressure and mixing ratio conversions, reversible entropy, and density temperature; we also include less common expressions which pyPI relies on or makes use of: CAPE, the minimum pressure estimate, the empirical LCL equation, PI efficiency, and the potential intensity decomposition.

2. We have sought to highlight the many assumptions in pyPI by (for the first time, in this study) fully documenting in section 3 the two algorithms which form the pyPI code base.

In response with the other reviewer, we considered the effects of several specific assumptions (both numerical and scientific). We discussed how changes in pLCL or CAPE could affect PI calculations, how the minimum pressure convergence threshold might affect the influence PI values, and the role of dissipative heating.

The adjustable parameters define the existing set of assumptions which can be quickly tested by users without any further code modifications. A (non-exhaustive) list of additional assumptions that would require code changes to address includes the definition of the outflow temperature, the LCL definition, inclusion of Ck/CD variability as a function of wind speed, and inclusion of a tropospheric stratification factor. Alternative characterizations of PI—such as the ocean coupled index (Lin et al. 2013) or "surface

PI" (Rousseau-Rizzi and Emanuel 2019)—could also be included in the repository as additional separate functions to improve pyPI's utility and flexibility.

We have added a short discussion in section 4 on these existing assumptions and opportunities for improvement, including the future inclusion of a tropospheric stratification factor (1-s(Gamma)) suggested by Kieu and Wang (2017). The section synthesizes what we see as the primary opportunities for growth in the pyPI.

There are certainly more assumptions which could be addressed than we mention here. For individual planned improvements in pyPI, we make use of the Projects tool in the Git repository to inform the community of our goals. We also strongly encourage users to bring them to our attention using the repository's "Issues" tool or by directly contacting the developer.

Minor comments

Eq. 14: Thank you for noting this typo. This should be R_d, and "Rd" has now been replaced in equation 14 with the appropriate "R_d" constant. Eq. 14 has also been updated for improved readability, in response to comments from the other reviewer.

References: Lin, I. I., Black, P., Price, J. F., Yang, C. Y., Chen, S. S., Lien, C. C., Harr, P., Chi, N. H., Wu, C. C., and D'Asaro, E. A. (2013), An ocean coupling potential intensity index for tropical cyclones, Geophys. Res. Lett., 40, 1878-1882, doi:10.1002/grl.50091.

Rousseau-Rizzi, R., & Emanuel, K. (2019). An Evaluation of Hurricane Superintensity in Axisymmetric Numerical Models, Journal of the Atmospheric Sciences, 76(6), 1697-1708.